# Spatial-Temporal Consistency Enhanced Segmentation for Laparoscopic Surgical Videos

**Ewen Rondel**[*1]                                          EWENRONDEL@GMAIL.COM
**Lin Guo**[*1]                                               LIN.GUO@SLU.EDU
[1] *Computer Science Department, Saint Louis University, USA*

**Mohammad Mahmoud**[2]                                      MMAHMOUD96@SIUMED.EDU
[2] *School of Medicine, Southern Illinois University, USA*

**Flavio Esposito**[1]                                        FLAVIO.ESPOSITO@SLU.EDU

**Editors:** Accepted for publication at MIDL 2025

## Abstract

Accurate segmentation of laparoscopic surgical videos is essential for enhancing intraoperative guidance and improving patient outcomes. However, this task remains challenging due to the constrained field of view, visual clutter, frequent occlusions, and inconsistent illumination. To address these challenges, we propose SSTC-Seg (Surgical Spatial-Temporal Consistency Segmentation), a lightweight deep learning framework for video-based segmentation. It integrates a memory system and a Hierarchical Dense Conditional Random Field (HD-CRF) with skip connections for spatial details preservation to refine coarse predictions and model contextual relationships across frames. Evaluated on the Dresden Surgical Anatomy Dataset (DSAD), SSTC-Seg achieves competitive multi-organ segmentation performance with significantly fewer parameters compared to existing state-of-the-art methods.

**Keywords:** Computer Vision, Segmentation, CRF, Machine learning.

## 1. Introduction

Integrating advanced technologies into laparoscopic surgery offers considerable potential to improve procedural accuracy, operational efficiency, and patient outcomes. Among these innovations, computer vision has emerged as a transformative tool, enabling real-time video segmentation to support intraoperative decision-making and reduce the risk of human error (Maier-Hein et al., 2017; Twinanda et al., 2016). This capability is particularly valuable in robotic-assisted surgery, where novice surgeons often struggle with spatial orientation and organ recognition due to limited sensory feedback and complex anatomical environments.

Despite advances in vision-based segmentation, most existing approaches are either developed for natural images with wide fields of view (Kirillov et al., 2023; Yang et al., 2018) or tailored for static medical imaging modalities such as MRI or CT (Akkus et al., 2017; Ronneberger et al., 2015). These methods often fail to generalize to the unique visual domain of laparoscopic procedures. Surgical videos introduce specific challenges, including a narrow and dynamically shifting field of view, frequent occlusions and visual clutter from instruments and tissue manipulation, proximity to organ boundaries that obscure anatomical context, and highly variable illumination with specular reflections. These factors make accurate segmentation particularly difficult.

---

[*] Contributed equally

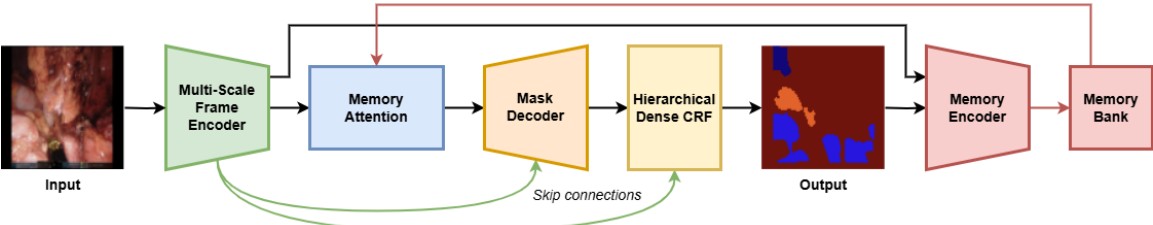

Figure 1: Overview of the proposed SSTC-Seg architecture. HD-CRF refines the coarse segmentation mask. Green lines represent skip connections that provide hierarchical features for improved spatial awareness. The refined output is encoded and stored in a memory bank, which feeds back into the memory attention module via the red loop to promote temporal consistency across frames.

Given these challenges, it is essential to design segmentation frameworks that explicitly model both spatial and temporal consistency. Maintaining spatial coherence ensures anatomically meaningful boundaries, while temporal consistency is crucial for robust performance across video frames in real-time surgical environments. To address these issues, we propose SSTC-Seg (Surgical Spatial-Temporal Consistency Segmentation). Architecturally, SSTC-Seg adopts an encoder-decoder backbone similar to U-Net, but modified for multi-scale detection, and extends it with dense CRF modules to explicitly model inter-object and intra-object relationships, resulting in sharper segmentation boundaries and improved contextual understanding.

We validate our approach on the Dresden Surgical Anatomy Dataset (DSAD) (Carstens et al., 2023), covering eight anatomical classes. Both quantitative and qualitative evaluations demonstrate that our method achieves performance comparable to state-of-the-art techniques while maintaining a lightweight parameter structure.

## 2. Method

Our proposed SSTC-Seg architecture builds on a dual-branch encoder-decoder backbone, enhanced with a memory bank for temporal modeling, U-Net-style skip connections for spatial detail preservation (Ronneberger et al., 2015), and a Hierarchical Dense Conditional Random Field (HD-CRF) for structured refinement (Zhang et al., 2015; Ladickỳ et al., 2013). An overview of the complete network structure is shown in Fig. 1.

The HD-CRF module, applied after the mask decoder, refines coarse predictions by modeling spatial and temporal relationships through dense pairwise connections and contextual features from skip connections. This promotes anatomical boundary sharpness and local consistency, particularly in the presence of occlusion and lighting artifacts. A multi-scale refinement strategy further improves robustness: a downsampled version of the prediction undergoes parallel HD-CRF processing, then is upsampled and fused with the full-resolution output, enhancing accuracy in complex surgical scenes.

Finally, the memory bank encodes and updates historical feature representations, allowing the network to leverage temporal context across frames. This mechanism enhances temporal consistency and improves the model's ability to track dynamically changing anatomical structures over time (Zhu et al., 2024).

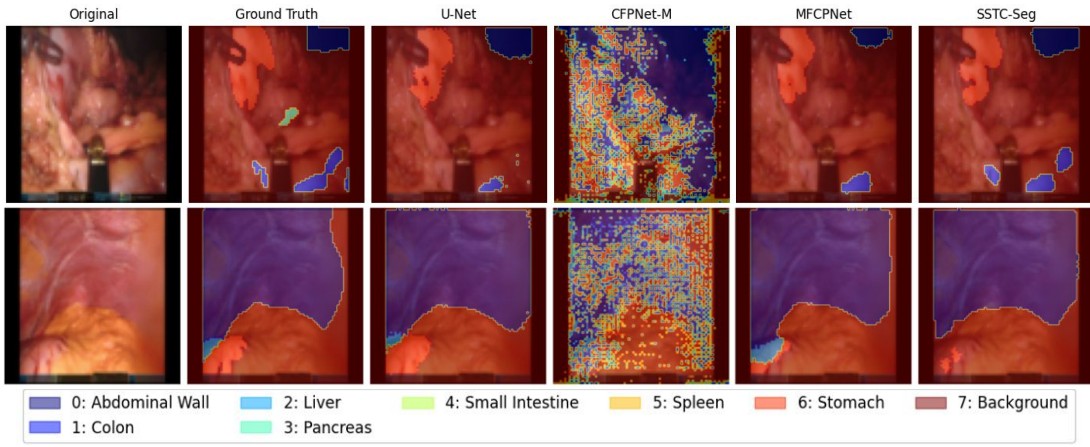

Figure 2: Qualitative comparisons.

## 3. Experimental Results

We built published methods including UNet (Ronneberger et al., 2015), CFPNet-M (Lou et al., 2023), and MFCPNet (Hou et al., 2025) for comparison. All models were adapted to support multi-class segmentation. Both qualitative (Fig.2) and quantitative results (Table.1) demonstrate that SSTC-Seg achieves performance on par with state-of-the-art methods such as MFCPNet, while using significantly fewer parameters (10M vs. 45M), underscoring its computational efficiency. Moreover, SSTC-Seg achieves the lowest Hausdorff Distance among all compared methods, indicating superior boundary localization—an essential attribute for precise surgical guidance. Compared to the U-Net baseline, our method exhibits consistent improvements across all metrics. While CFPNet-M has shown strength in general medical image segmentation, its performance degrades significantly in this setting, highlighting the unique difficulty of laparoscopic multi-class segmentation tasks and the importance of tailored solutions like SSTC-Seg.

Table 1: Model-wise Performance Comparison

| Method | Params | Accuracy(%) | Dice | Jaccard | HD95 |
|---|---|---|---|---|---|
| UNET | 31.0M | 80.3 | 0.628 | 0.615 | 11.133 |
| CFPNet-M | 0.54M | 41.3 | 0.139 | 0.092 | 33.899 |
| MFCPNet | 45.7M | 84.87 | 0.712 | 0.699 | 8.39 |
| **SSTC-Seg (Ours)** | 10.0M | 82.17 | 0.6786 | 0.6492 | 7.6838 |

## 4. Conclusion

This study introduces SSTC-Seg, a lightweight yet effective segmentation framework designed to address the challenges of multi-organ segmentation in laparoscopic surgical videos. By integrating a memory-guided attention mechanism, U-Net-style skip connections, and a Hierarchical Dense Conditional Random Field (HD-CRF) module, our method promotes both spatial and temporal consistency, resulting in improved segmentation accuracy in visually complex surgical environments. In future work, we plan to further develop our method by refining the network architecture and improving memory modeling strategies. We also aim to evaluate SSTC-Seg on larger-scale datasets with more anatomical classes to further validate its generalizability and robustness in diverse surgical scenarios.

## Acknowledgments

This work has been supported by SIU School of Medicine and by NSF award OAC 2430236.

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
