# OpenReview forum: "Spatial-Temporal Consistency Enhanced Segmentation for Laparoscopic Surgical videos"
_MIDL.io/2025/Short_Papers — MIDL 2025 - Short Papers_

### Official Review · Reviewer_VLmZ · 2025-04-25

**Rating:** 3
**Confidence:** 4

**Summary:**

The manuscript presents a method for surgical video segmetnation that leverages a memory module to contend with challenges in surgical videos, such occlusion.

**Strengths:**

+ Increasing spatial consistency in video segmentation/processing is of interest

+ A lightweight model might be useful

**Weaknesses:**

- The focus of the manuscript is on temporal segmentation but the evaluation does not focus on this; defaulting to the standard frame-wise metrics. What experiments can be performed to demonstrate benefits of this approach for the video?

- The baselines seem weak; recent segmetnation models such as SAM2 (maybe not directly applicable here, still) already use memory to better deal with video segmentation. Stronger baselines would considerably strengthen the evdience presented here.

---

### Decision · Program_Chairs · 2025-05-01

Accept